# Bioactive Lignans from Flaxseed: Biological Properties and Patented Recovery Technologies

**Paola Sangiorgio** [1,*] **, Simona Errico** [1] **, Alessandra Verardi** [1] **, Stefania Moliterni** [1] **, Gabriella Tamasi** [2] **, Claudio Rossi** [2] **and Roberto Balducchi** [1]

1 ENEA, Italian National Agency for New Technologies, Energy and Sustainable Economic Development, Trisaia Research Centre, 75026 Rotondella, MT, Italy
2 Department of Biotechnology, Chemistry and Pharmacy, University of Siena, Via Aldo Moro 2, 53100 Siena, SI, Italy
* Correspondence: paola.sangiorgio@enea.it

**Abstract:** Flaxseed lignans frequently feature in the literature. However, much remains to be discovered about the mechanisms underlying their functional and therapeutic properties. Furthermore, it is necessary to identify systems for lignan production and detoxification that are sustainable, cost-effective, easy to use, and scale up. These systems can address the needs of the nutraceutical, cosmetic, and pharmaceutical sectors and lead to competitive commercial products. This review analyzes the biological effects of lignans as anticancer, antioxidants, and modulators of estrogen activity. It also focuses on the most recent articles on lignan extraction methods that are sustainable and suitable as products for human consumption. Furthermore, the most up-to-date and relevant patents for lignan recovery are examined. The search and selection methodology for articles and patents was conducted using the most popular bibliographic and patent databases (e.g., Scopus, Pubmed, Espacenet). To the best of our knowledge, this is the first overview that details the patented technologies developed in the flaxseed lignans area in the last 10 years.

**Keywords:** flaxseed; lignans; secoisolariciresinol diglucoside (SDG); SDG functional properties; solvent extraction; flaxseed lignan patents





## 1. Introduction

Flax (*Linum usitatissimum*) is an annual herbaceous plant with blue flowers, widespread in various parts of the world. Its Latin name means "very useful" and, throughout human history, this plant has been used for very different purposes, as a fiber plant and oil crop. The flax seed consists of an embryo, two cotyledons, and a hull, in the approximate proportions of 4%, 55%, and 36%, respectively. Flaxseed hulls can vary in color from yellow to brown. The difference in pigment does not involve variations in proximate composition, functional properties, or biological activities [1].

Flaxseeds are rich in fats, particularly polyunsaturated fatty acids (around 70% of total fats). Linolenic acid, an omega-3 fatty acid with beneficial health properties, is the prevalent polyunsaturated fatty acid [2].

The chemical composition of flax seeds depends mainly on the cultivar, harvest time, geographical origin and post-harvest processing method [3,4]. Most of the lipids and proteins are concentrated in the cotyledons, while the flaxseed carbohydrates are condensed in the hulls [5]. As for micronutrients, flax seeds have a high amount of minerals (especially potassium), vitamins E and B3 [2], and phenolic compounds [6].

Flaxseeds are of great interest to the food industries because of their functional properties [7]. They give particular sensory characteristics to foods, when used as ingredients during processing, storage, and use as additives [6,8]. Adding flaxseeds to packaged products such as bread, cookies, bars, soups, and snacks has been shown to confer functional properties to these products, increase their shelf-life, and enhance sensory characteristics

valued by consumers. Flaxseed addition, however, should not exceed 20% so as not to adversely affect the texture, normal gluten behavior, and overall acceptability by consumers who do not like an excessive nutty taste in such products [9].

Regarding flaxseed oil, it has excellent functional properties, as it slows down the oxidation process and prevents rancidity in products to which it is added, but it is used in moderation because it is easily prone to oxidative degeneration [10].

Of great interest is flaxseed mucilage or gum, known for its probiotic qualities and high capacity to bind water and retain moisture. These characteristics make it an ideal ingredient to increase the consistency, viscosity, and stability of some beverages and stabilize some pork-based foods [11]. It is used more than other food gums because less is needed to achieve the same degree of texture in the food [12].

However, alongside the nutritional and beneficial compounds, flaxseeds have compounds with toxic or antinutritional effects. There are cyanogenic compounds with potential toxicity for humans and animals and antinutrients such as linatine and phytic acid [12,13].

Flaxseeds are one of the plant matrices richest in lignans. Lignans are a heterogeneous group of molecules found in almost all higher plants [14] belonging to the phytoestrogen family, which also includes other molecules such as isoflavones, coumestans, and flavonoids [15]. Lignans play a role in the defense of the plant and seeds against diseases, pathogens, and herbivores [16].

Chemically, although the structural model may differ, they are formed by two phenylpropane units (C6-C3) linked by their carbon 8 with a β-β′ bond, so they are generically referred to as diphenolic compounds or phenylpropanoid dimers [14,17]. These phenylpropane units are often called "monolignols": the most common are the p-coumaryl, coniferyl, and synapyl alcohols, which differ in the methoxylation on the aromatic ring [14]. In particular, coniferyl alcohol is found predominantly in the cell walls of plant tissues [18]. The polymerization of monolignols gives rise to natural lignin, a molecule characteristic of woody plants [14].

Various types of lignans can be formed depending on the bonding between different phenylpropane units. Based on the cyclic structure, carbon skeleton, and oxygen position, eight groups of lignans have been identified: furofuran, furan, dibenzylbutyrolactone, dibenzylbutane, dibenzylcycloctadiene, dibenzylbutyrolactone, arylnaphthalene, and aryltetralin [17].

Lignans are molecules widely distributed throughout the plant kingdom, although some plants synthesize higher amounts in different parts of the plant (roots and seeds). Cereals, fruits, and vegetables contain low levels of lignans, while flax and sesame seeds have higher concentrations, especially flax seeds. It has been estimated that the amount of lignans present in flaxseed is 75 to 800 times higher than in cereals, legumes, fruits, and vegetables [3].

The applications of lignans are numerous thanks to their beneficial and nutritional properties as antioxidant, anti-inflammatory, and anticancer agents. They are used in the food industry as ingredients in functional foods and beverages or as food supplements [19]. They enter as constituents in cosmetics and toiletries, such as for hair or skin care [20]. Lignans are also used as active ingredients in pharmaceutical products [21,22].

The global market for lignans is projected to reach over 90 million US Dollar by 2026 [23]. Diseases due to inadequate lifestyles are progressively increasing, and therefore it is foreseeable that consumer demand for food supplements, in particular, flaxseed, will increase, especially in light of their numerous beneficial effects on human health [16].

Several articles have been published in recent years on flaxseed and the lignans they contain. Our review, in addition to briefly considering the biological effects of flaxseed lignans, analyzes the extraction methods developed so far, which must necessarily consider the difficulty of recovering positive bioactive molecules on a large scale without, however, extracting and concentrating antinutrients, such as cyanogenic glycosides. This aspect is especially crucial for supplement and nutraceutical manufacturers, who need to apply these methods in practice. Furthermore, to highlight the great attention these compounds

have received, in this review, we give an updated overview of the principal patents in the sector. In fact, the last scientific article by Hosseinian and Beta, which analyzes the patented techniques for SDG lignans' extraction and isolation from flax seeds, dates back to 2009.

Finally, we compare the technologies used in the patents published in the last ten years with those of the scientific articles of the same period to highlight the trends and the constraint points.

The literature search for this review was carried out using the databases of PubMed, Scopus, and Google Scholar, with the following key terms: "lignans", "secoisolariciresinol", "biological functions", "therapeutic properties", "extraction", "isolation", "recovery", and "flaxseed" in various combinations depending on the specific review section. We focused on scientific studies from 2000 to today. Only peer-reviewed articles written in English were included in the search list. The list was then skimmed by reading the title and the abstract. Only eligible manuscripts have undergone full-text evaluation. The most pertinent articles served to find further bibliographic references. Section 4.2 reports in detail the search methodology used for patents. The literature and patent search was conducted on November 2022.

## 2. Biological Properties of Lignans

Lignans are, as mentioned, phenolic compounds and, as such, have a strong antioxidant capacity, which inextricably links them to beneficial effects on human health [5]. Moreover, secoisolariciresinol (SECO), like the other lignans found in flaxseed (matairesinol, lariciresinol, and pinoresinol), are mammalian estrogen precursors that are converted to enterolignans, enterodiol, and enterolactone (Figure 1), by the anaerobic intestinal microflora [24].

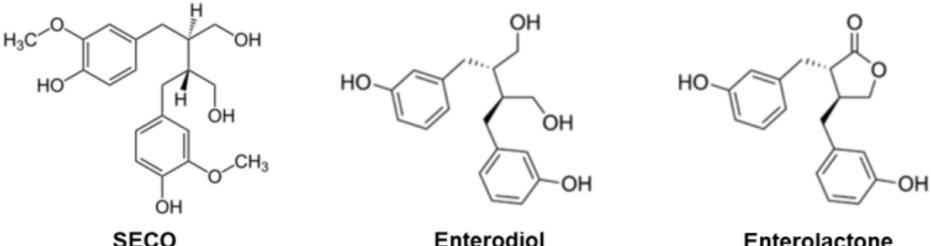

**Figure 1.** Secoisolariciresinol (SECO), precursor of enterodiol and enterolactone.

Lignans are now recognized as anticholesterol [25–27], antiviral [28], anticancer [29,30], antioxidant [7,31], supplemental for improved athletic performance [32], antidiabetic [33,34], estrogenic and anti-estrogenic [35,36], anti-inflammatory [37], anti-depressant [38,39], antibacterial, and anti-fungal [40]. Figure 2 summarizes the main beneficial effects of lignans.

Due to their phenolic characteristics, lignans play an essential role as "scavengers" of hydroxyl radicals, which makes them a valuable defense against the development of diseases caused by the free radicals produced by the human body during the oxidation of fats, proteins, and carbohydrates. Free radicals damage tissues, membrane lipids, nucleic acids, and proteins and can cause cancer, lung disease, neurological disease, premature ageing, and diabetes [3].

Due to their similarity to mammalian estrogens, lignans can help treat cancers related to hormone metabolism [18]. They can bind to estrogen receptors, altering the functionality of estrogens, in particular, lowering their circulation in the bloodstream and their biological activity, thus reducing the risk of developing cancer [3]. Their mechanism of action would actually appear to be much more complex, as they can, for example, influence intracellular enzymes and protein synthesis, stimulate the production of sex hormone-binding globulin in the liver, and reduce the concentration of free hormones in plasma. They can also interact with sex steroid-binding proteins and act as inhibitors of several steroid-metabolizing enzymes, thus playing a protective role against breast and colon cancers [14]. In the specific case of breast cancer, it has been suggested that secoisolariciresinol diglucoside (SDG) may

play a protective role due to its ability to regulate the expression level of zinc transporters (zinc concentration is higher in breast cancer cells than in normal cells). In addition, one of the metabolites of SDG, enterolactone, would be able to suppress the proliferation, migration, and metastasis of cancer cells [41].

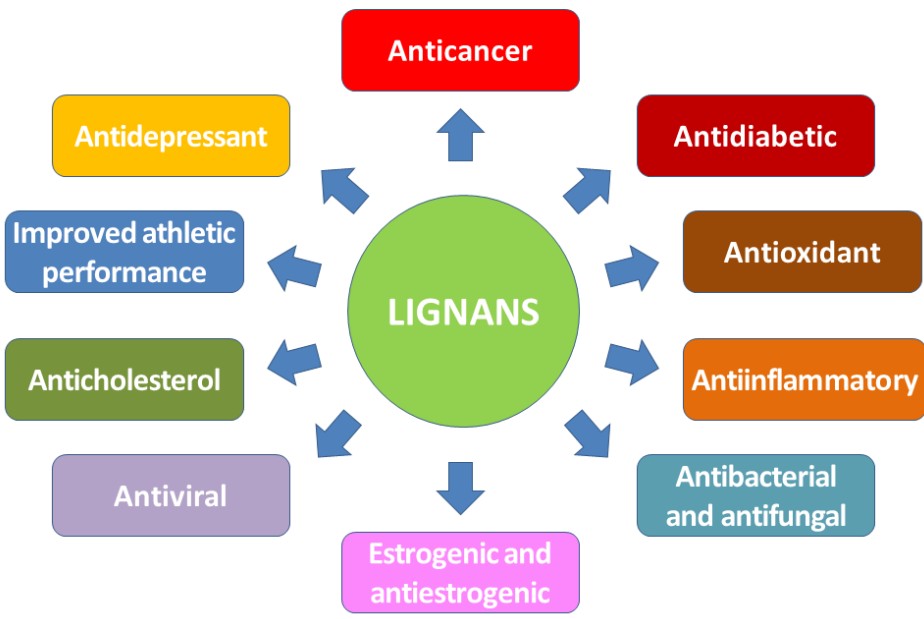

**Figure 2.** Beneficial properties of lignans.

Specifically, lignans inhibit aromatase activity in adipose tissue of obese postmenopausal women. The result is evident in the reduction of circulating estrogen and serum levels of sex steroid hormones, which are implicated in the development of breast cancer [42].

Due to their antioxidant capabilities, lignans (particularly SDG) help reduce the risk of lupus nephritis, oxidative DNA damage, and lipid peroxidation, as well as prevent oxidative stress associated with metabolic syndrome [43].

A relevant activity of flaxseed (particularly its oil, rich in ALA, lignans, and fibers) is lowering cholesterol. Lignans, acting as selective estrogen receptor modulators, can reduce LDL levels (the "bad" cholesterol) and triglycerides and normalize HDL cholesterol in the blood [26]. In this way, flaxseed lignans highly reduce the risk of cardiovascular disease. In particular, SDG can prevent or delay the progression of atherosclerosis and thus prevent coronary artery disease, stroke, and peripheral arterial vascular disease [44].

Lignans also have antiglycemic activity, inhibiting the development of type 1 and type 2 diabetes by lowering the glycemic response in the blood [33]. Lignans' antidiabetic activity is added to the influence of flaxseed fibers on insulin secretion and plasma glucose homeostasis. In fact, SDG reduces the concentration of C-reactive protein, which is related to insulin resistance in type 2 diabetes and glucosuria, so it may help reduce the incidence of type 1 diabetes and delay the development of type 2 diabetes in humans [43]. In addition, by prolonging satiety, lignans actively help reduce obesity [5].

Other recognized beneficial effects on human health include immunomodulatory activity, anti-leishmaniosis activity, inhibition of 5-lipoxygenase, and high efficiency in protection against rheumatoid arthritis. Since they can also bind to tubulins of the cytoskeleton, they can play an important antiviral action by interfering in the replication of viruses. In addition, they inhibit reverse transcriptase, so they are also effective in blocking the replication of RNA viruses [17].

Lignans can also have a role in reducing the symptoms of depression and stress: during these times, the human body produces pro-inflammatory cytokines (such as TNF- and IFN-) that cause mood swings. Lignans' intake appears to promote the production of

two polyunsaturated fatty acids-EPA (eicosapentaenoic acid) and DHA (docosahexaenoic acid), which counteract such mood swings [1].

SDG has shown effects on mental stress. SDG administration to ovariectomized mice inhibited stress-induced behavioral changes and reversed the increase in serum corticosterone and adrenocorticotropic hormone levels induced by chronic stress [38]. Moreover, the effects of three flax cultivars with different lignan content were studied on postmenopausal women with vascular disease who showed signs of stress due to having to perform frustrating cognitive tasks. In these women, the intake of lignans significantly reduced blood pressure during the period of stress; in particular, the cultivar with the highest lignan content increased plasma fibrinogen levels much less than the others and reduced plasma cortisol levels more [45].

Given the strong binding of lignans to estrogen, their potential effect on the reproductive system has been studied and found to be influential. Although further studies are needed, in the meantime, attention should be paid to flaxseed consumption during pregnancy and lactation. Early results are comforting in that the intake of lignans during lactation reduced susceptibility to mammary carcinogenesis later in life, with no adverse effects on selective reproductive indices in mothers or offspring [43].

Despite the wealth of this information and the knowledge of so many mechanisms involving lignans, many studies are still necessary to answer still-unanswered questions (toxicity, bioavailability) and to better understand the role of lignans in health and disease prevention. In this way, it will be possible to develop adequate dietary guidelines and nutraceutical or pharmaceutical products capable of preventing cardiovascular disease and helping fight cancer [1]. To this end, it will be necessary to set up and optimize lignan extraction, purification, and analysis systems in order to have considerable quantities of pure lignans for in vitro and in vivo investigations and clinical studies [46].

## 3. Flaxseed: A Source of Lignans

In addition to containing high levels of polyunsaturated fats and proteins (see Table 1), flax seeds are one of the principal sources of lignans.

**Table 1.** Chemical composition of flaxseed [2].

| Name | Amount (%) |
|------|------------|
| Water | 6.96 |
| Protein | 18.29 |
| Total fat | 42.16 |
|    Saturated Fatty Acids | 3.66 |
|    Monounsaturated Fatty Acids | 7.52 |
|    Polyunsaturated Fatty Acids | 28.73 |
|       Linolenic acid | 22.81 |
|       Linoleic acid | 5.90 |
| Carbohydrate | 28.88 |
|    Fiber, total dietary | 27.30 |
|    Sugars, Total | 1.55 |
| Others (minerals, vitamins, carotenoids, lignans, etc.) | 3.71 |

In flaxseed, lignans are particularly abundant in the seed coat [5]; the hull alone contains higher amounts of lignans than the whole seeds, suggesting that their biosynthesis occurs mainly here [43]. Of the lignans present, the most abundant is SDG (610–1300 mg per 100 g of flaxseed), but other lignans, such as matairesinol, pinoresinol, and lariciresinol, although present in lesser amounts, are certainly valuable as well [3,5].

Lignans biosynthesis in plants starts from the coniferyl alcohol. Two units of this precursor are coupled in the presence of dirigent proteins (DIR) to form pinoresinol which is sequentially reduced to form lariciresinol and then SECO. SECO is dehydrogenated to form matairesinol. Figure 3 illustrates the biosynthesis of the major lignans in flaxseed and the catalysts involved in the reactions (adapted from [37]).

**Figure 3.** Biosynthesis of the major lignans in flaxseed. NGT (pinoresinol glucosyltransferase), PLR (pinoresinol/lariciresinol reductase), LGT (lariciresinol glycosyltransferase), SGT (secoisolariciresinol glycosyltransferase), SID (matairesinol O-methyltransferase), Glc (Glucoside). Adapted from [37].

Table 2 shows the amounts of the major lignans present in flaxseed as reported by the Phenol-explorer database [47].

**Table 2.** Amounts of the major lignans in flaxseed [47].

| Lignan | Amount (mg) per 100 g of Flaxseed |
|---|---|
| Secoisolariciresinol (SECO) | 257.60 |
| Lariciresinol | 11.46 |
| Pinoresinol | 8.64 |
| Matairesinol | 6.68 |

SECO is mainly present in its glycosylated form (SDG) and constitutes 95% of the lignans present in flaxseed [17]. Unlike other lignans that are present in free form, the SDG in flaxseed is bound in an oligomeric structure called the lignan complex [43]. SDG molecules are bound via 3-hydroxy-3-methylglugaric acid (HMGA) in a biopolymer weighing about 4000 Da (of which SDG constitutes about 35% by weight), consisting of five SDG units and four HMGA units [18]. In addition to SDG and HMGA, the hydroxycinnamic acids, coumaric acid glucoside, ferulic acid glucoside, and herbacetin diglucoside have been reported as constituents of the lignan complex [21].

For this reason, the high variability in the measured quantities of SDG and SECO in flaxseed (as well as in other matrices) is due not only to the cultivar, geographic origin, and year of harvest but also to the type of treatment the matrix undergoes before analyses. The extraction method, indeed, has a relevant role in the lignans' yield. Alkaline and acid treatments under strong conditions can be destructive or lead to other lignans [48].

Therefore, a lignans recovery process is fundamental for manufacturers of supplements or nutraceuticals to maximize the yields of lignans and, at the same time, prevent the extraction and concentration of antinutrient or toxic compounds [49].

In addition to beneficial compounds, flax seeds, in fact, contain antinutritional compounds, such as linatine, cyanogenic glycosides, and phytic acid [3]. The scientific literature does not report any adverse effects related to the direct consumption of flax seeds. However, these antinutrients could become harmful at high levels once concentrated in the extraction

phases of bioactive compounds from flaxseeds or in the flaxseed cake after oil removal [5]. Furthermore, their presence compromises the bioavailability and bioaccessibility of the nutritional value of flaxseeds. For this reason, it is crucial to resort to detoxification techniques that can break or remove these antinutrients [12].

## 4. Recovery Technologies of Lignans from Flaxseed

Many extraction methods for the recovery of natural products from plant matrices have been studied with the advancement of chemical technologies. Organic solvent extraction remains the most used method. By optimizing the extraction parameters, such as solvent polarity, solvent–solid ratio, particle size, extraction time, and temperature, it is possible to increase the yield of the process [46].

However, traditional solvent extraction processes using Soxhlet, maceration, and digestion have several disadvantages: (1) they use large volumes of solvents with associated chemical hazards; (2) they take a lot of time and energy; (3) they commonly use high temperatures to accelerate extraction, which can degrade heat-sensitive compounds; (4) they can co-extract and concentrate other undesired components. New efficient processes are now available to overcome these drawbacks, in the framework of the so-called green extraction methods, such as ultrasound-assisted, microwave-assisted, enzyme-assisted, supercritical fluid, and high-pressure extraction [50].

In any case, the obtained extract must go through purification processes to remove unwanted compounds and retrieve lignans of high purity [51].

Lastly, the whole process should be user-friendly, cost-effective, environmentally sustainable, and scalable so as to meet the needs of industrial production, lead to competitive commercial products, and achieve the goals of the European Green Deal [52].

Regarding flaxseed lignans, the most studied source in the literature is the defatted meal or the flaxseed oilcake, but ground hulls and whole flaxseed are also taken into consideration. The most studied lignan is SDG, followed by its aglycone SECO [53].

The following two sections illustrate the lignan recovery methods reported in the literature in the last twenty years and the patented processes of the last decade.

### 4.1. Recovery Technologies from the Scientific Literature

As mentioned, in flaxseed, SDG is esterified with HMGA and other phenolic compounds (see Section 3) to form a macromolecular complex. Therefore, unbound SDG is commonly obtained by means of sequential extraction of the lignan macromolecule, followed by acid, alkaline, and enzymatic hydrolysis to degrade the lignan complex [54]. Acid hydrolysis effectively breaks both the ester and glycoside bonds, while alkaline hydrolysis breaks only ester bonds. However, under strong acid conditions, the released SECO can convert to anhydrosecoisolariciresinol (AHS) [55]. On the other hand, direct alkaline hydrolysis carried out under particular conditions can lead, in a reasonable time, to a higher SDG yield than that obtained from hydrolysis after alcoholic extraction [56]. Recently, a research study has proposed a one-pot reaction using ammonium hydroxide in aqueous ethanol to liberate and extract SDG from flax hulls [57].

Over time, scientific studies have focused on optimizing lignan recovery processes by suitably varying the extraction solvents according to the lignan structure. High-polarity lignans such as SDG require polar solvents. The literature reports the use of solvents such as ethanol + 1,4-dioxane (1:1), ethanol, methanol, acetone, isopropanol, and butanol [58].

Table 3 shows different processes used for the lignan extraction from flaxseed matrices with the related method or conditions and results. As shown in the Table, several articles use microwaves [59–61], ultrasound [62,63], or pulsed electric fields [63] to improve solvent extraction. Other scientists, on the other hand, use solvents in particular state conditions, such as supercritical fluids [64] or subcritical water [65,66].

**Table 3.** Different processes for lignan extraction from flaxseed reported in the literature since 2003.

| Extraction Techniques | Raw Materials | Extracted Lignans | Operating Conditions | Results | Ref. |
|---|---|---|---|---|---|
| Cellulase-assisted extraction | Whole flaxseed, Flax hull | SDG | Sequential methanol/ethanol extraction followed by alkaline hydrolysis and subsequently enzyme-assisted extraction of SECO | SDG: 40.75 mg/g in the hull; 15.20 mg/g in whole flaxseed. Cellulase is more efficient than β-glucosidase | [54] |
| Direct alkaline hydrolysis | Milled defatted flaxseed flour | SDG | Direct hydrolysis with 5 mL of 2 M NaOH for 1 h at 20 °C. | SDG yields are higher than those obtained by hydrolysis after alcoholic extraction | [56] |
| One-pot hydrolysis and extraction | Flaxseed hull | SDG | Alcoholic ammonium hydroxide (pH = 12.9) for 4.9 h at 75.3 °C allows the direct hydrolysis and extraction of SDG. | Very simple method with very high yields of SDG. | [57] |
| Microwave-assistedExtraction (MAE) | Flaxseed | SDG | 70% ethanol added with 0.1 M NaOH. 50–150 W power for 1–15 min | MAE has higher yields of SDG (16.1 mg/g) than that of traditional methods. | [59] |
| | Flaxseed hull | SDG | 0–100% ethanol, liquid to solid ratio from 5:1 to 40:1 mL/g, 50–390 W microwave energy for 10–330 s | SDG recovery with MAE (11.7 mg/g) is higher than that of Soxhlet extraction (7.6 mg/g) | [60] |
| | Defatted flaxseed meal | SDG | 1 g sample extracted with 50 mL 0.5 M NaOH, with the microwave power (135 W) applied intermittently for 3 min | 97% SDG recovery. Fast and efficient method. | [61] |
| Ultrasound Assisted Extraction (UAE) | Flaxseeds | SDG | Water with 0.2 N NaOH to get free SDG from its HMG complex. Extraction: 60 min at 25 °C and 30 kHz ultrasound frequency | Higher yield for SDG, particularly compared to MAE and conventional alkaline extraction | [62] |
| | Ground defatted flaxseeds | SDG | 1. UAE with methanol/$H_2O$ (75/25, *v/v*).<br>2. UAE with 0.08 M NaOH/methanol on the residue.<br>3. Extracts from (1) and (2) are combined and hydrolyzed by 0.02 M NaOH.<br>4. Acidification and analyses by UPLC/ESI-HRMS. | No purification step is necessary before analysis.Efficient recovery of SDG. | [63] |
| Supercritical $CO_2$ Extraction | Ground (hydrolyzed) flaxseed, ground hulls | SDG | Extraction of SDG by using SC-$CO_2$ modified with 7.8 mol% ethanol at 60 °C and 45 MPa | Very low yields compared to the original SDG content. SC-$CO_2$ extraction can be an advantageous pretreatment. | [64] |
| Subcritical water extraction | Defatted flaxseed meal | SDG | Extraction using subcritical water at 170 °C, at pH 9 and 5.2 MPa, and solvent to solid ratio of 100 mL/g | Simultaneous extraction of SDG (21 mg/g), proteins, and carbohydrates. | [65] |
| | Flaxseed meal sticks | SDG | High SDG yields at 170–180 °C for 15 min, 1.500 psi and 40% fresh water | SDG: 72.57% (at 180 °C), 70.67% (at 170 °C) | [66] |
| High voltage Electric Discharge (HVED) | Flaxseed cake | SDG | HVED treatment at 20–40 kV and 20–60 °C, 0–25% ethanol for subsequent extraction | Lower levels of SDG extracted compared to the literature data | [67] |

**Table 3.** *Cont.*

| Extraction Techniques | Raw Materials | Extracted Lignans | Operating Conditions | Results | Ref. |
|---|---|---|---|---|---|
| Pulsed electric fields (PEF) | Flaxseed hull | Polyphenols (lignans, flavonoids, ferulic, and p-coumaric acids) | PEF treatment (at 20 kV/cm for 10 ms) after hull rehydration using water, ethanol (20%), and 0.3 mol/L NaOH for 40 min at 20 °C. | PEF extracts uo to 80% of polyphenols (including lignans) | [68] |
| Negative-pressure cavitation extraction | Flaxseed cakes | SDG | Extraction at −0.04 MPa, at 35 °C, for 35 min with ethanol 65% (*v/v*), ventilation volume 90 L/h, NaOH 1.39%, and the liquid/solid rate (mL/g) 13.16:1. | SDG yields (16.25 mg/g) and SDG extraction purity (3.86%) are comparable to UAE results. | [69] |

It is important to note that the choice of solvent and the extraction conditions strongly determine the biological activity of the extracts. Moreover, the extraction method can affect the levels of antinutrients in flaxseed extracts. Ethanol or methanol extraction can concentrate cyanogenic glycosides to toxic levels [49]. On the contrary, aqueous extraction strongly reduces their concentration to acceptably safe levels [13].

Furthermore, the extraction can be affected by the presence of fatty acids. It is for this reason that, very often, the starting source of lignan is the defatted meals or linseed cakes, which have undergone the extraction of the lipophilic content with a non-polar solvent such as petroleum ether and n-hexane [46].

*4.2. Patented Recovery Technologies*

The patent search was carried out on 3 November 2022 through the advanced search mode of Espacenet, the European Patent Officer [70]. A search for <lignans> OR <lignan>, by choosing in the Text fields the <title or abstract>, produces 1026 results, from 1964 to the present. Of these 1026 patents, 155 have a publication date prior to 2000, while 524 are the results patented in the last ten years, demonstrating the ferment of activity around lignans. If we search for <lignans> OR <lignan>OR< secoisolariciresinol > in the <title or abstract>, we reach 1103 results from 1964 to date. If we further narrow down the above search by adding the keywords <flaxseed> OR <linseed> OR <flax> in the <title or abstract>, we obtain 138 patents since 1995. Of these 138, 70 patents use lignans or SDG as ingredients to obtain various products with different functions, such as increasing athletic performance, treating neurological and viral diseases, preventing diabetes, supplementing food, etc. In addition, 26 patents concern various food preparations (oils, beverages, biscuits, dietary fibers, etc.), as well as cultivation methods for lignin accumulation and technologies for the comprehensive utilization of flaxseed. The remaining 42 patents deal with methods for extracting lignans from flaxseed and detoxifying flaxseed extracts or products. Figure 4 schematizes the patent search methodology on the Espacenet database and related results.

Table 4 lists the 26 patents published in the last decade, from the oldest to the most recent. The link to the publication number allows direct consultation of the patents present in the Espacenet database [70].

**Figure 4.** Scheme of the patent search methodology and the related results.

**Table 4.** Patents concerning extraction methods of lignans from flax matrices from 2012 to present [70].

| Title | Flaxseed Materials | Extracted Lignans | Number and Ref. |
|---|---|---|---|
| Method for extracting secoisolariciresinol diglucoside from flax seeds or husks, extract obtained and use thereof | Flax seeds or husks | SDG | CN101570556B [71] |
| Production process of secoisolariciresinol diglucoside | Flaxseed oil residue | SDG | CN102558252A [72] |
| Method for extracting, separating, and purifying flax lignans from flax cakes | Flaxseed cake | Not specified: SECO or SDG | CN102796148A [73] |
| Method for extracting secoisolariciresinol diglucoside | Flaxseed cake | SDG | CN102816190A [74] |
| Detoxification method of flaxseed product | (Defatted) flaxseed powder | Not specified | CN102823782A [75] |
| Extraction of lignans from flaxseeds through ultrasonic enzymolysis | Defatted flaxseeds | Not specified | CN102125261B [76] |
| Method for isolation of flaxseed lignans | Flaxseed cake | Not specified | PL397907A1 [77] |
| Method for directly extracting flax lignans from flaxseed meal and detoxifying high-protein flaxseed meal | Flaxseed meal | SDG | CN104478954A [78] |
| Method for continuously extracting flaxseed gum and lignan from flaxseed husks | Flaxseed husks | SDG | CN102850414B [79] |
| Method for extracting lignan from linseed oil residue | Flaxseed oil residue | Not specified | CN102766174B [80] |
| Method for preparing fruit antioxidant by efficiently extracting flax lignans | Flaxseeds | Not specified | CN105145806A [81] |
| Separation and purification method for secoisolariciresinol diglucoside | Flax seeds or flax shells | SDG | CN103396461B [82] |
| Method for preparing flax lignans from flax cake | Flax cake | Not specified | CN105585599A [83] |

**Table 4.** *Cont.*

| Title | Flaxseed Materials | Extracted Lignans | Number and Ref. |
|---|---|---|---|
| Method for extracting secoisolariciresinol diglucoside, obtained extract and use of extract | Flaxseed oil residue | SDG | CN103860649B [84] |
| Method for obtaining high biological activity lignan and mushroom by using edible mushroom to degrade flax seed shell | Flax seed shell | SDG | CN105000934B [85] |
| Extraction method for flax lignan from flax seed husks | Flaxseed husks | Not specified | CN108129526A [86] |
| Method for continuously extracting flaxseed gum and secoisolariciresinol diglucoside from flaxseed meal | Flaxseed meal | SDG | CN108409813A [87] |
| Method for extracting flax lignans from flaxseed husks by microwaves | Flaxseed husks | SDG | CN108659064A [88] |
| Extraction method of flax gum and lignan in flax cake | Flax cake | Not specified | CN109535276A [89] |
| Lignan extraction technology | Flaxseeds | Not specified | CN111116687A [90] |
| High-performance liquid phase method for simultaneously preparing and separating four lignan components | Flaxseed hulls or meals | SDG, SG, SECO, AHS | CN111233944A [91] |
| Extraction method of flaxseed extract | Flaxseeds | Not specified | CN111543644A [92] |
| Method for extracting secoisolariciresinol diglucoside from flaxseeds by using subcritical water | Flaxseeds powder | SDG | CN112480192A [93] |
| Method for simultaneously producing five components in flaxseeds | Flaxseeds | SDG | CN113527385A [94] |
| Method for removing benzopyrene in flax extract | Flaxseed meal | SDG | CN113980066A [95] |
| Subcritical composite solvent extraction separation method for multiple components in flaxseed meal | Flaxseed meal | Not specified | CN113786641B [96] |

Abbreviations: SDG: secoisolariciresinol diglucoside; SG: secoisolariciresinol glucoside; SECO: secoisolariciresinol; AHS: anhydrosecoisolariciresinol.

A previous review, now outdated, had described some processes of extraction/isolation of SDG from flaxseed, patented in the 1998–2006 period [97].

Since that period, patents have increasingly focused on the use of non-traditional technologies to assist solvent extraction, such as microwaves [71,81,88,90], ultrasound [71,73,74,76,85,86,91], supercritical fluids [81], and subcritical solvents [93,96]. The most commonly used solvents are ethanol and water in various concentrations and combinations, sometimes mixed with sodium hydroxide [79].

In the most recent patent, Li et al. proposes a subcritical compound solvent extraction and separation method using n-butane and anhydrous ethanol (1:1 ratio) at 69 °C for 2 h [96]. The obtained supernatant is the crude lignan extract, containing lignan polymers which undergo alkali treatment with 0.1 M NaOH for at least 24 h to release SDG. This subcritical extraction allows the separation of flaxseed lignans and oils in one step.

Huang et al. patented in 2021 a process for SDG extraction from flaxseed using subcritical water [93]. The extraction process involves a temperature of 100–200 °C for 10–80 min and an optimal pressure range of 3–6 MPa.

Several inventions describe multi-step processes for separating hulls from kernels, gum from proteins, or oil from the lignan-rich fraction [89,92]. The latter are mostly extracted using ethanol, sometimes in combination with ultrasound [76]. For example, a recent invention describes a method for producing five components from flaxseed: oil, gum, SDG, protein, and flaxseed cyclic peptide [94]. This method does not use toxic solvents (only water, ethanol, and sodium hydroxide) and temperatures above 60–80 °C, so it does

not degrade nutrients. The inventors declare that their process is simple, sustainable, and cost-effective.

A high-efficiency liquid phase method is described for the simultaneous preparation and separation of SDG, secoisolarociresinol glucoside (SG), SECO, and anidrosecoisolari-ciresinol (AHS) [91]. The process begins with the hydrolysis (ultrasound dissolution with water for 1–20 min), followed by acidolysis (1–5 h at 80–120 °C). The subsequent step of liquid chromatography preparation culminates with the elution from a reverse phase C18 column using a gradient with an aqueous solution of acetonitrile.

A simple method of making flax lignans from flax cake uses ethanol for leaching the flax cake powder [83]. The temperature is 45–65 °C, the ethanol concentration is 65–80%, and the leaching time is 5–7 h. The extract is filtered with an ultrafiltration membrane and then concentrated using a nanofiltration membrane. After adding NaOH to the concentrate, a large amount of colloidal precipitate is obtained. The liquid fraction, adjusted to a pH of 6.5, is adsorbed by the AB-8 macroporous resin (at an adsorption temperature of 15–25 °C). Elution is carried out sequentially with 20% and 30% ethanol solution, and the combined eluate was concentrated and dried to obtain flax lignans. A similar process scheme is used by another patent which, however, uses different ethanol concentrations, extraction times, and temperatures [84].

Lastly, several patents use acid treatment with edible acids to remove cyanogenic glycosides from flax matrices. Acetic or citric acids are the most preferred. After stirring for 3–6 h at room temperature, the sample thus treated is heated in a microwave oven (power: 700 W, 2450 MHz) or in an oven. Such a treatment has been shown to reduce cyanogenic compounds without degrading lignans [75].

### 4.3. Comparison between the Literature Results and Patents of the Last Decade

The previous subsections have shown the extraction technologies of lignans from flax matrixes most used in the scientific literature (Section 4.1) and in patents (Section 4.2) in the last twenty years.

The comparison and merging of the research results in the literature and the patent database of the last decade lead to noteworthy results (Table 5). First, we have only nine articles describing methods of extracting lignans from flax materials from 2012 onwards. In the same period, there were 26 patents, a much higher number than articles in the literature. One explanation could be that the preliminary research phases for filing a patent begin long before its publication date. However, even if we extended the literature search for another ten years (since 2003), the number of articles would rise to 15, always remaining lower than the 26 patents of the last ten years.

**Table 5.** Extraction technologies of lignans from flax matrices: comparison between the literature results and patents in the last decade.

| Extraction Technologies | Results in the Literature of the Last Decade (Total Number 9) | Patents of the Last Decade (Total Number 26) |
|---|---|---|
| Conventional solvent extraction | 4 | 12 |
| Microwave-assisted Extraction (MAE) | 0 | 4 |
| Ultrasound Assisted Extraction (UAE) | 2 | 7 |
| Supercritical $CO_2$ Extraction | 0 | 1 |
| Subcritical solvent extraction | 1 | 2 |
| Electrical methods (PEF and HVED) | 2 | 0 |

Secondly, ultrasound and microwave-assisted extraction technologies are in good numbers among the patents (seven and four, respectively). Three patents use extraction solvents in the subcritical and supercritical states. No article deals with this subject. On the

contrary, electric technologies such as High voltage Electric Discharge (HVED) or Pulsed electric fields (PEF) are not present among the patents, thus remaining only at a scientific research level.

Another relevant consideration is the high number of patents using conventional chemical technologies (12 patents). They are not particularly innovative overall but try to respond to the need to provide a simple method at a large scale to obtain more products in addition to lignans, such as flaxseed oil, gum, and proteins. Moreover, these patents want to combine various steps to properly pretreat the flax matrices before extracting lignans and, at the same time, purify and detoxify the extracts. Conventional technologies are also present in the literature (four articles) but mostly describe the optimization of small-scale extraction processes.

Finally, practically all the inventors claim to have found the solution for producing pure lignans, with high yields, on an industrial scale (or with easily scalable processes) simply and inexpensively. The demonstration of this will be in the commercial field. For now, lignans extracted from flaxseed (but also from other matrices) remain high-priced for high-value commodities [23]. Therefore, to reach the broader market, the industry will need technologies for lignan extraction and recovery that are advantageous in terms of environmental and economic sustainability and possibly lead to a cascade of high-value products.

## 5. Conclusions

Due to their health-beneficial compounds, flaxseeds have attracted interest in the last decade, especially in the functional and vegan food sector.

The flaxseed lignan industry is booming due to the therapeutic potential of flaxseed in the treatment of high-incidence chronic diseases in industrialized countries, such as hypertension, diabetes, and cardiovascular and neurological diseases. Furthermore, flaxseed lignans can be successfully used as nutraceutical ingredients or incorporated into functional products to combat diseases associated with an unhealthy lifestyle by adopting as natural a strategy as possible.

In the last 20 years, research has led to the development of numerous technologies for the extraction and detoxification of lignans from flaxseed, and to multiple patented inventions. However, to date, an up-to-date overview of the state of the art in this area was lacking.

This review has filled this gap, using a search and selection methodology based on high quality bibliographic and patent databases.

The results of this research show that the solvent extraction of flaxseed lignans is more efficient if supported by unconventional technologies, such as ultrasound and microwave. In this case, it is possible to use green or low toxic solvents, such as water and ethanol.

However, further studies are needed to develop more sustainable, efficient, and effective recovery methods for obtaining purified and detoxified lignans at a large scale. In this way, it will be possible to further explore the biological activities of lignans and respond to the growing market demand.

**Author Contributions:** Conceptualization, P.S., S.E., and A.V.; methodology, P.S.; writing—original draft preparation, P.S., S.E., and S.M.; writing—review and editing, P.S., S.E., A.V., and G.T.; visualization, P.S. and S.M.; supervision, R.B. and C.R. All authors have read and agreed to the published version of the manuscript.

**Funding:** This research received no external funding.

**Institutional Review Board Statement:** Not applicable.

**Informed Consent Statement:** Not applicable.

**Data Availability Statement:** Not applicable.

**Conflicts of Interest:** The authors declare no conflict of interest.

## Abbreviations

| | |
|---|---|
| AHS | anhydrosecoisolariciresinol |
| DIR | dirigent proteins |
| Glc | Glucoside |
| HMGA | 3-hydroxy-3-methylglugaric acid |
| HVED | High voltage Electric Discharge |
| LGT | lariciresinol glycosyltransferase |
| MAE | Microwave-assisted extraction |
| NGT | pinoresinol glucosyltransferase) |
| PEF | Pulsed electric fields |
| PLR | pinoresinol/lariciresinol reductase |
| SDG | secoisolariciresinol diglucoside |
| SECO | secoisolariciresinol |
| SG | secoisolariciresinol glucoside |
| SGT | secoisolariciresinol glycosyltransferase |
| SID | matairesinol O-methyltransferase |
| UAE | Ultrasound Assisted Extraction |

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
