# Peer review of "Bioactive Lignans from Flaxseed: Biological Properties and Patented Recovery Technologies"

_nutraceuticals, doi:10.3390/nutraceuticals3010005_

Round 1

Reviewer 1 Report

Comments and Suggestions for Authors

Major comments:

1.      Title: The title needs revision;  patented bracket is not needed as the paper elaborates on patents.

2.      Abstract: The abstract is written well. However, elaboration from which databases the literature was collected.  The conclusion in the abstract is not clear.

3.      Section 1: Introduction: The introduction is very concise. The background and rationale of the study are written very well written.

Though authors have added a section on how literature was collected however how data was retrieved and extracted, how many papers were browsed, excluded paper and included paper flow chart need to be added, and which software was used for retrieval of data. A rearrangement of the sections is required to understand the manuscript.

4.      Section 2. Biological properties of lignan

Figure 2. page 3, Authors may check the spelling such as anti-oxidant or antioxidant, anti-bacterial or antibacterial, etc.

5.      Overall, one figure showing how bioactive molecules have pharmacological activities may be depicted.

6.      Section 4.2.

Table 4. Rearrange the table for example inventors, title, raw materials, Number and reference. The reference column can be kept separate.

7.      In a new heading on the Research gap and future recommendations can be included.

8.      Section 5. The conclusion needs revision and concluding remarks of the review may be added.

9.       Overall grammatical and English editing is required in the manuscript.

Author Response

We have offered detailed responses to the comments of reviewer 1 in the attached file

Reviewer 2 Report

Comments and Suggestions for Authors

It is an interesting review paper, long and quite exhaustive at some point, I would want to see more schemes and figures in order to follow all the documented data.

I believe it is useful but I recommend to make some modifications:

- please make the methods based either on the lignan type or based on the flax matrix, it is combined, mixed, difficult to make an overview of the recory method for example for SECO;

- I do not believe the list of inventors must be pointed out, it will be on the reference list, I would appreciate to see the extracted lignan, for example;

- the same for Table 3, re-organize based in defined parameters, mixed, difficult to follow;

- I would add a scheme with the most used methods found in the literature, then based on patents, a comparasion;

- not clear what was the point with the presentation of both platforms, since no comments on that;

- perspective is general;

- also, what is the contribution of the authors on this field, if the authors wrote a review paper on lignans, this means they are working with lignans; 

- probably a perspective based on their findings in comparation with the literature. 

Author Response

We have offered detailed responses to the comments of reviewer 2 in the attached file 

Round 2

Reviewer 1 Report

Comments and Suggestions for Authors

The authors have added all the reviewer comments and justified their paper. The paper may be accepted.

Author Response

Thank you for your appreciation and for helping us improve our manuscript.

Reviewer 2 Report

Comments and Suggestions for Authors

The authors improved the manuscript, but there are some minor issues to be resolved:

- the abstract: "To our knowledge, the last review focused on flaxseed lignan patents dates back to 2009. Our review is therefore helpful to give an updated overview of the patented technologies developed in this area  in the last 10 years. ", I would reformulate, "To the best of our knowledge, this is the first overview that details the patented technologies developed in flaxseed lignans area  in the last 10 years.". and describe the review from 2009 in the introduction, in order to point out the upgraded text, and so on.

- an abbreviation list at the end of the manuscript.

Author Response

Thanks to your suggestions, we further improved our manuscript:

  • we have changed the abstract and described the review of the patents from 2009 in the introduction;
  • we have added an abbreviation list at the end of the manuscript.